



# True polar wander and heat flux patterns at the core-mantle boundary in a mantle convection simulation

Thomas Frasson[1], Stéphane Labrosse[2], Henri-Claude Nataf[1], and Nicolas Coltice[3]

[1]Univ. Grenoble Alpes, Univ. Savoie Mont Blanc, CNRS, IRD, Univ. Gustave Eiffel, ISTerre, 38000 Grenoble, France
[2]ENS de Lyon, Lyon, France
[3]ENS, Paris, France

**Correspondence:** Thomas Frasson (thomas.frasson@univ-grenoble-alpes.fr)

**Abstract.** The core-mantle boundary (CMB) heat flux is an important variable of Earth's thermal evolution and dynamics. Seismic tomography enables access to seismic heterogeneities in the lower mantle, which can be related to present-day thermal heterogeneities. Alternatively, mantle convection models can be used to either infer the past CMB heat flux or to produce statistically realistic CMB heat flux distributions in self-consistent models. Mantle dynamics modifies the inertia tensor of the
Earth, which implies a rotation of the Earth with respect to its rotation axis called True Polar Wander (TPW). This rotation has to be taken into account if mantle dynamics is to be linked to core dynamics. In this study, we explore the TPW and the CMB heat flux produced by a self-consistent mantle convection model. The geoid is also computed and investigated in order to determine the driving mechanism of TPW. This model includes continents, dense chemical piles at the bottom of the shell and plate-like behavior, providing the possibility to link TPW and the CMB heat flux with plate tectonics. A principal
component analysis (PCA) of the CMB heat flux is computed to obtain the dominant heat flux patterns. The model shows a geoid dominated by upper mantle structures. Subduction zones and continents are correlated with positive geoid anomalies, about 20 times larger than the observed geoid anomalies. Chemical piles are mostly correlated with negative geoid anomalies because of the anti-correlation between the positions of subducting slabs and the piles. TPW thus tends to lock continents and subduction zones close to the equator, while chemical piles are shifted towards higher latitudes. The positive CMB heat flux
anomalies are mostly located at low latitudes because of the equatorial slabs. The dominant heat flux patterns obtained by the PCA largely reflect the supercontinent cycle captured by the model, providing CMB heat flux patterns representative of the supercontinent formation and dispersal.

## 1 Introduction

The heat flux at the core-mantle boundary (CMB) is an important variable of the Earth's thermal evolution and dynamics.
The CMB heat flux notably plays a key role in the dynamics of the core and the geodynamo. The mean value of the CMB heat flux controls the cooling rate of the core, and thus the power available for dynamo action. Both the mean value and the lateral variations of the CMB heat flux affect the behavior of the geodynamo in numerical simulations, with strong effects on the magnetic reversal frequency and on the angle between the rotation axis and the axis of the magnetic dipole (Glatzmaier et al., 1999; Kutzner and Christensen, 2004; Olson et al., 2010). Large heat flux heterogeneities can even prevent dynamo



action (Olson and Christensen, 2002). It is therefore important to evaluate what could be the evolution of the CMB heat flux on geologic timescales, in order to assess the consequences for the geodynamo.

The viscosity in the Earth's mantle being much larger than in the Earth's outer core, the mantle sees the CMB as an isothermal boundary, while the core sees the CMB as an imposed laterally varying heat flux. This heat flux changes on mantle convection time scales much larger than those relevant to the dynamics of the core. The level of knowledge of the CMB heat

flux, and notably its spatial distribution, depends on our understanding of the lower mantle structure and dynamics. Seismic tomography offers a view of the lower part of the mantle, revealing more and more complex structures (e.g. Dziewonski et al., 1977; Lay and Helmberger, 1983; Garnero and Helmberger, 1995; Su and Dziewonski, 1997; Durand et al., 2017) (Ritsema and Lekić, 2020, for a review). Near the equator, two antipodal large low shear velocity provinces (LLSVPs) interpreted as thermochemical piles are particularly well defined at the bottom of the mantle in those models (Garnero and McNamara, 2008).

They form a characteristic structure dominated by degree 2 spherical harmonics. The LLSVPs are correlated with the degree two geoid, positive geoid anomalies being found over the two piles (Dziewonski et al., 1977). At smaller scales not seen in global tomographic models, heterogeneities like the ultra low velocity zones (ULVZs) have also been observed using dedicated approaches (e.g. Rost et al., 2005). Despite these improvements, it is still difficult to have a clear view of the CMB heat flux pattern. Thermal and chemical effects are difficult to separate in the tomographic models (Trampert, 2004; Mosca et al., 2012),

while only providing a snapshot of the Earth's history. The reconstruction of the eruption sites of hotspots suggests that the LLSVPs could have stayed fixed for the past 300 Myr at least (Burke et al., 2008), providing a stable large-scale heat flux pattern through time. This view of stable LLSVPs has however been challenged by recent seismic tomography models (Davaille and Romanowicz, 2020) and mantle flow reconstructions (Flament et al., 2022). Estimates of the past CMB heat flux have been obtained for the last 450 Myr from reconstructions of the mantle flow driven by observed plate motions (Zhang and Zhong,

2011; Olson et al., 2015). These models show that the CMB heat flux pattern is governed by plate motion through subducting slabs, which cool the lower mantle. Large chemical piles at the base of the mantle are also found to reduce the heat transfer, thereby increasing the overall lateral heterogeneities of CMB heat flux. These CMB heat flux reconstructions can then be used to constrain the evolution of the core and of the geodynamo. Olson et al. (2015) use the results of these models to drive a thermal evolution model of the core, while Zhang and Zhong (2011) find equatorial heat flux minima around 170 Myr and 100

Myr ago, which coincide with the Kiaman superchron and Cretaceous superchron, respectively. This kind of model is however limited by the accuracy of plate reconstructions, which remain poorly constrained before the Pangea assembly (Müller et al., 2022). Alternatively, estimates of the CMB heat flux can be obtained by self-consistent models, without prescribed surface velocities (Liu and Zhong, 2015; Coltice et al., 2019). This approach is less "Earth-like" than the models with prescribed plate motion in the sense that it does not aim at reproducing the Earth's past. It however enables to obtain some statistically realistic

information concerning the mantle convection depending on input parameters. Nakagawa and Tackley (2008) notably showed using this kind of model that the lateral variations of the CMB heat flux could be as large as the mean.

This work aims at describing the CMB heat flux produced by a realistic mantle convection simulation that exhibits plate-like behavior, and at providing representative CMB heat flux maps to be used in geodynamo simulations. In contrast with previous studies (Zhang and Zhong, 2011; Olson et al., 2015), the mantle model used in this study is self-consistent, without prescribed





surface velocities. The representative CMB heat flux maps are obtained using a Principal Component Analysis (PCA) to bring out the dominant heat flux patterns at the CMB.

Earth's rotation plays a crucial role in the liquid core. If we are to explore the impact of realistic CMB heat flux patterns on the geodynamo, it is essential that these patterns be produced in a reference frame that preserves the axis of rotation. The mantle convection simulation does not depend upon the position of the Earth's spin axis since the rotational forces are 65 negligible for the mantle, and the boundary conditions at the surface and at the CMB are not affected by a global rotation of the mantle with respect to its spin axis. However, the mass redistribution and the boundary topographies caused by convection modify the moments of inertia of the mantle (Munk and MacDonald, 1960; Phillips et al., 2009), which can be obtained from the degree 2 coefficients of the geoid (MacCullagh, 1845; Schaber et al., 2009). The mantle therefore rotates in order to keep its axis of greatest inertia along the rotation axis (Goldreich and Toomre, 1969). This is called the True Polar Wander 70 (TPW). The TPW emerging from mantle convection models has been studied to retrieve the past track of the rotation axis (Steinberger and O'Connell, 1997; Schaber et al., 2009), or to investigate the link between supercontinents and TPW in freely-evolving convection models (Zhong et al., 2007; Phillips et al., 2009). In these models, plate tectonics is either prescribed or approximated. Here, we compute the TPW occurring in a model reproducing plate-like behavior and rotate the simulation frame accordingly, before computing the CMB heat flux patterns.

Section 2 describes the methods used for the different steps of the analysis. The results are presented in Sect. 3 and discussed in Sect. 4.

## 2 Methods

### 2.1 3D mantle convection model

Our study rests upon an analysis of the mantle convection simulation obtained by Coltice et al. (2019), focusing on the CMB 80 heat flux and the geoid. This simulation computed 1131 Myr of mantle evolution using the StagYY code in a 3-D Yin-Yang geometry (Tackley, 2008) at a high Rayleigh number, $\mathrm{Ra} = 10^7$. The model state is saved every million years and we extract CMB heat flux and the non-hydrostatic geoid anomaly maps at each time-step. This model has been set up to reproduce Earth-like mantle convection features, with particular attention to plate-like behavior. Figure 1 gives a sketch of the mantle structure at the beginning of the simulation. Continents are modeled as a material whose composition differences from normal 85 mantle cause it to be less dense and more viscous, with 200 km-thick interiors and 125 km-thick rims. At the base of the mantle, two antipodal, initially 500 km-thick chemical piles are introduced to model the Atlantic and Pacific LLSVPs with a denser and more viscous material than the surrounding mantle. The reference density $\rho_0$ and the reference viscosity $\eta_0$ used in the model have dimensional values of $4000 \mathrm{~kg~m^{-3}}$ and $10^{22}$ Pa s, respectively. The compositional density deficit inside continents is $\delta\rho_c = 225 \mathrm{~kg~m^{-3}}$, and the compositional density excess in the basal piles is $\delta\rho_p = 137 \mathrm{~kg~m^{-3}}$. The initial state 90 of the simulation is an equilibrated mantle circulation obtained with fixed chemical piles and fixed continents assembled in a Pangea-like supercontinent placed above the Atlantic pile. At the start of the simulation, continents and piles are left free to move. The relaxation to a new statistically steady state takes about 300 Myr.



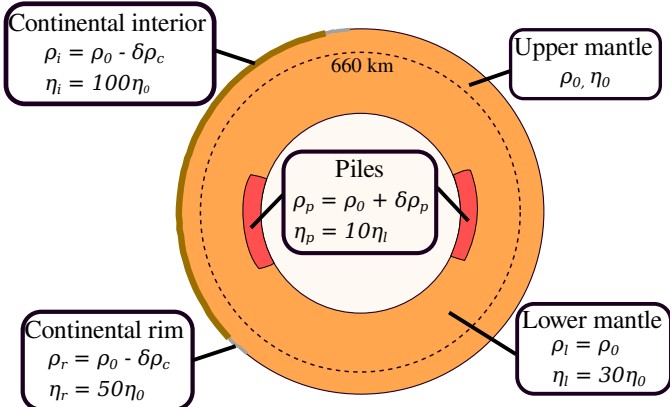

**Figure 1.** Sketch of the mantle structure at the beginning of the simulation. The reference density and viscosity of the lower mantle ($\rho_l$, $\eta_l$), the piles ($\rho_p$, $\eta_p$), the continental interior ($\rho_i$, $\eta_i$), and the continental rims ($\rho_r$, $\eta_r$) are given as a function of the reference density and viscosity of the upper mantle ($\rho_0$, $\eta_0$). The density anomalies $\delta\rho_c$ and $\delta\rho_p$ account for the lower density of continents and higher density of the piles, respectively.

Plate-like behavior is obtained using a pseudoplastic rheology and a temperature-dependent viscosity (Tackley, 2000a, b). An uppermost 14 km-thick weak layer in oceanic regions allows for asymmetric subduction zones (Crameri et al., 2012). This model reproduces a statistically realistic mantle convection, with global features (plate velocities, surface heat flux, hypsography, plume buoyancy flux) as well as local features (continental breakup, rifting, back-arc extension, mantle plumes) that fit the observations (Coltice et al., 2019). Thanks to the extreme temperature-dependence of viscosity in the model, plumes notably display kinematic, thermal and buoyancy properties similar to those of the Earth (Arnould et al., 2020). Because plumes carry information from the CMB, such properties of the model are fundamental for our study.

Regarding the objective of this work, the Earth-likeness of surface processes and the long time evolution are the main advantages of this mantle convection simulation. It captures the effect of realistic plate tectonics on a CMB heat flux varying laterally and with time over a long time series. The model notably contains a complete cycle of formation and breakup of a super-continent, which is thought to modulate the CMB heat flux (Olson et al., 2013; Amit et al., 2015). The complexity of the CMB region being less well understood, the similarity between the model and the Earth at the bottom of the model is less certain. The use of the chemical piles nevertheless allows for strong, large scale temperature heterogeneities, as they are revealed by seismic tomography (Trampert, 2004; Mosca et al., 2012).

## 2.2 Geoid

The time-evolution of the geoid controls the True Polar Wander. The geoid is an equipotential of gravity at the surface of the model. It can be computed by integration of the lateral density variations across the mantle model. Additional contributions arise from lateral mass heterogeneities produced by the deflection of the interfaces, in particular at the surface and at the CMB. These deflections are not explicit in the model (which assumes spherical boundaries), but they can be computed from the $\tau_{zz}$





element of the stress tensor $\boldsymbol{\tau}$ at the interfaces, where $z$ denotes the vertical coordinate. Geoid calculation is an intrinsic capacity of the StagYY code. It follows Zhang and Christensen (1993) and is implemented in a similar fashion as in Zhong et al. (2008). The topography and geoid anomalies are decomposed into spherical harmonic components on both interfaces, accounting for

self-gravitation effects as an effective pressure term. Using the flow solver, it separately computes for each harmonic degree the surface geoid anomalies and topographies corrected for self-gravitation. Because of the large viscosity lateral variations, this procedure is required over simpler methods based on geoid kernels assuming radial viscosity distributions (Richards and Hager, 1984; Ricard et al., 1984). We refer to Zhong et al. (2008) for a more detailed description of the method. The spherical harmonic coefficients of the geoid anomalies $c_{l,m}$ and $s_{l,m}$ are computed at each time step in the simulation. The geoid $N(\lambda, \phi)$

can then be expressed as a function of latitude $\lambda$ and longitude $\phi$ as:

$$N(\lambda,\phi) = R \sum_l \sum_m \left[\, c_{l,m} \cos\, m\phi + s_{l,m} \sin\, m\phi\, \right] P_l^m(\sin\lambda), \tag{1}$$

with $R$ the radius of the Earth, and $P_l^m$ the associated Legendre polynomial of degree $l$ and order $m$.

## 2.3 True Polar Wander implementation

TPW is governed by the conservation of the Earth's angular momentum, yielding Liouville's equation (Ricard et al., 1993).

Here we use a simplified approach to compute the TPW by considering that the Earth's spin axis aligns instantaneously with the principal axis of greatest inertia (Steinberger and O'Connell, 1997; Zhong et al., 2007). Those principal axes are obtained through the computation of the geoid perturbation induced by the modeled mantle convection. The inertia tensor $\mathbf{I}$ due to the mass redistribution in the mantle is built from the coefficients of the degree 2 geoid, following MacCullagh's formula (MacCullagh, 1845; Schaber et al., 2009):

$$\mathbf{I} = MR^2 \sqrt{\frac{5}{3}} \begin{pmatrix} \dfrac{c_{2,0}}{\sqrt{3}} - c_{2,2} & -s_{2,2} & -c_{2,1} \\[2mm] -s_{2,2} & \dfrac{c_{2,0}}{\sqrt{3}} + c_{2,2} & -s_{2,1} \\[2mm] -c_{2,1} & -s_{2,1} & -2\dfrac{c_{2,0}}{\sqrt{3}} \end{pmatrix} \tag{2}$$

where $M$ and $R$ are the mass and radius of the Earth, respectively. The principal inertia axes are then obtained through a diagonalization of this matrix. The axis of greatest inertia corresponds to the highest eigenvalue, while the two equatorial principal axes correspond to the smallest and intermediate eigenvalues. Computing the axis of greatest inertia gives two new poles, one on each side of the axis. Which one of the two poles is the "north pole" is arbitrary and it is chosen at the beginning

of the simulation. TPW is then implemented iteratively by rotating the mantle at each time step to align the axis of rotation with the axis of greatest inertia, following the meridian which contains both axes. The rotation direction is chosen so that the new north pole remains in the same hemisphere as the previous one, effectively limiting TPW's amplitude to 90° per iteration (the time step between two iterations being 1 Myr, this corresponds to a maximum velocity of 90° Myr$^{-1}$).

This TPW implementation corresponds to a change in the reference frame in which the data are represented. This new frame

is permanently wandering with respect to the initial simulation frame, we thus call it the wandering frame in the following.





The spherical harmonic transforms as well as the rotations to correct for the TPW are performed using the SHTns library (Schaeffer, 2013). SHTns provides an efficient implementation of spherical harmonic rotations based on the stable recursive evaluation of Wigner's d-matrix proposed by Gumerov and Duraiswami (2015), which is accurate up to very large degrees $(> 10^4)$.

## 2.4 Principal Component Analysis of the heat flux at the CMB

We use a Principal Component Analysis (PCA) to obtain the dominant heat flux patterns at the bottom of the mantle. The PCA is a data analysis tool that can be applied to a data set comprising several observations, each observation depending on several variables. It is used to express the data set in a new orthonormal basis in order to limit the number of variables needed to explain the data. This is done by computing new variables (called principal components), which are combinations of the

initial variables. A full mathematical description of the PCA theory is given by Abdi and Williams (2010). See also Pais et al. (2015) for an application to core flows and details of the method. Considering a data set containing $I$ observations described by $J$ variables, the PCA consists in the singular value decomposition of the $I \times J$ data matrix $\mathbf{D}$ as

$$\mathbf{D} = \mathbf{WSP}, \tag{3}$$

where $\mathbf{W}$, $\mathbf{S}$ and $\mathbf{P}$ have respective dimensions $I \times K$, $K \times K$ and $K \times J$, with $K = \min(I, J)$ the rank of the data matrix. $\mathbf{P}$ is

a basis of $K$ new variables (or principal components), called $p_k$ with $k \in [\![1 ; K]\!]$, which are linear combinations of the initial $J$ variables. The amount of variance in the data set explained by $p_k$ decreases with increasing $k$. This variance is quantified by a score, called $s_k$, corresponding to the singular values in the diagonal matrix $\mathbf{S}$. The square of $s_k$ gives the variance explained by $p_k$. The projections of the $I$ observations on this new basis are stored in the $\mathbf{W}$ matrix.

In the framework of this study, PCA is used to obtain the principal components corresponding to heat flux patterns that

explain most of the heat flux signal at the CMB as a function of time. The data set consists in the spherical harmonics coefficients of $q_{CMB}$ for each snapshot of the mantle convection model, truncated at a maximum degree $l_{max} = 50$. To ensure that convection reached a statistical equilibrium, we removed the first 300 Myr of the simulation in the computation of the principal components. The data matrix thus contains I = 832 observations (the number of considered snapshots) and J = 2601 variables (the number of spherical harmonic coefficients). PCA requires data to be centered, which means in our case that the mean of

each spherical harmonic coefficient on the whole time series has to be removed. This operation is equivalent to removing the mean heat flux pattern from the time series. Following the notations described previously, the $k^{th}$ PCA component consists in a $J$-dimension vector $p_k(l, m)$ of spherical harmonic coefficients, a score called $s_k$, and a time-dependent weight $w_k(t)$. We note $\tilde{p}_k(\lambda, \phi)$ the heat flux pattern reconstructed from the $p_k(l, m)$ spherical harmonic coefficients. For each component, a time dependent amplitude $A_k$ can be defined as $A_k(t) = w_k(t) \times s_k$ with $t$ the time in the simulation. Because of the data centering,

the heat flux patterns of the PCA components have to be interpreted as perturbations to the average heat flux pattern, which will be called $\overline{q}(\lambda, \phi)$ in the following.





Once PCA is performed, the time-dependent CMB heat flux maps can be reconstructed as

$$q_{CMB}(\lambda, \phi, t) = \overline{q}(\lambda, \phi) + \sum_{k=1}^{K} w_k(t) s_k \tilde{p}_k(\lambda, \phi). \tag{4}$$

The first components give the highest contribution to the full variability of the CMB heat flux and can be used to interpret the
model in terms of mantle dynamics. They provide plausible CMB heat fluxes that can also be applied as boundary conditions
in dynamo calculations.

## 3  Results

### 3.1  Geoid

The geoid is computed in a self-consistent way in the mantle convection model. It should then be assessed whether the model
produces a geoid with Earth-like behavior. Figure 2 displays the evolution of the root mean square (RMS) of the geoid anomalies amplitude. The RMS lies between 550 m and 1200 m. For comparison, the observed non-hydrostatic geoid anomalies has

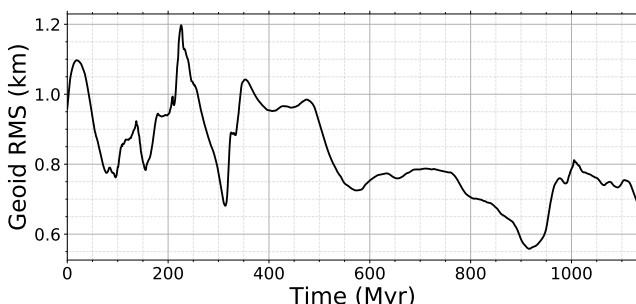

**Figure 2.** Time-evolution of the geoid RMS amplitude in the simulation.

an RMS amplitude of about 30 m (Hirt et al., 2013), more than one order of magnitude smaller than our computed geoid. The
geoid amplitude has however no effect on the position of the inertia axes, which is only controlled by the pattern of the degree
2 of the geoid.
Figure 3 displays the geoid, topography, and CMB heat flux maps for snapshots at 350 Myr, 600 Myr, 850 Myr and 1100 Myr.
The contours of the chemical piles at the CMB are highlighted on the heat flux maps. It can be seen from the four snapshots that
both continents and subductions are largely associated with geoid highs. The continents are generally associated with broad
positive anomalies, while subductions are associated with more local and stronger positive anomalies. Subductions often occur
at the edge of the continents, leading in some cases to strong geoid highs from the addition of both contributions. A correlation
between the chemical piles and the geoid is less obvious, though the piles are generally associated with geoid lows in the
displayed snapshots. Figure 4 shows the mean geoid anomalies in the areas where subductions, continents, or chemical piles
are located. Subduction zones are defined as the areas in which the temperature is below 80% of the mean temperature at 200



**Figure 3.** Maps of the geoid (left column), surface topography (center column) and $q_{CMB}$ (right column) in the mantle convection model after the TPW correction for snapshots taken at 350 Myr, 600 Myr, 850 Myr and 1100 Myr. The contours on the heat flux map follow the edges of the chemical piles. The inside of the piles is associated with low heat flux, while the outside is associated with high heat flux.




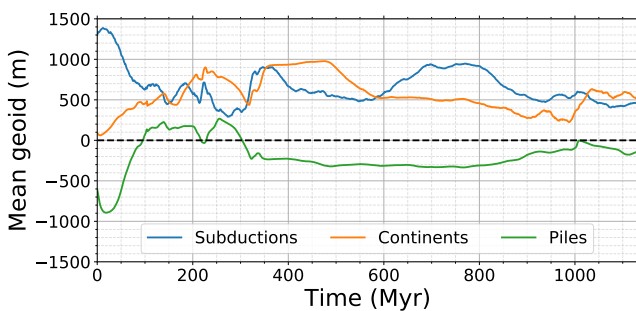

**Figure 4.** Evolution of the mean geoid anomalies over areas covered by subductions, continents and chemical piles during the simulation.

km depth. The continental areas are obtained using the surface composition field filtered up to spherical harmonic degree 20 in order to avoid considering the numerous micro-continents appearing at the end of the simulation. The geoid being computed

up to degree 20, the signal of these micro-continents cannot be well retrieved in the computed geoid. The piles area is taken at the CMB, corresponding to the contours shown in figure 3. The relationship between subduction zones and continents on the one hand, and geoid highs on the other, results in a positive mean geoid anomaly associated with continental and subduction zones. This mean positive anomaly is large for most of the simulations, lying mostly between 500 m and 1000 m for both subductions and continents. The mean geoid anomaly associated with chemical piles is lower, and changes sign four times

during the simulation. The piles are nonetheless associated with negative mean geoid anomalies for most of the simulation, especially between 320 Myr and 900 Myr. Except during the first tens of million years, the absolute value of the mean geoid anomaly on top of piles stays below 350 m. The correlation between the geoid and chemical piles seems thus low compared to the correlations with continents and subduction zones.

The TPW is controlled by the degree two of the geoid anomalies. Continents and subduction zones can thus be expected

to drive the TPW in the model. The TPW behavior, notably related to these plate-tectonic features as well as to the chemical piles, is the subject of the next section.

## 3.2  True Polar Wander

The TPW implementation consists of aligning the rotation axis with the axis of greatest inertia. This axis corresponds to the principal axis of the geoid having the lowest eigenvalue, which passes through the lows of the degree two geoid. TPW thus

tends to move the large-scale positive geoid anomalies at low latitudes. This is the case in the snapshots shown in Fig. 3, where high positive geoid anomalies are found close to the equator. Given the correlations shown Fig. 4 between the geoid and continents, subduction zones or chemical piles, TPW can be expected to affect the latitudinal distribution of these mantle features. Continents and subduction zones are positively correlated with the geoid, their latitudes should thus stay low in the wandering frame. The mostly negative correlation between geoid and chemical piles means that the piles can on the contrary

be expected to stay away from low latitudes. In the snapshots shown in Fig. 3, continents and subduction zones indeed appear





mostly at low latitudes, while chemical piles can be found at all latitudes but with spaces in between the piles occurring mostly near the equator. The effect of TPW on the latitudinal distribution of these features can be quantified by comparing the latitudes RMS of subduction zones, continents, or chemical piles in the wandering frame and in the simulation frame. We express this latitude RMS at a time $t$ as:

$$\lambda_x(t) = \sqrt{\int_{S_x(t)} \lambda^2 \frac{dS}{S_x(t)}}, \qquad (5)$$

where $x$ stands for subductions ($x = sub$), continents ($x = cont$) or chemical piles ($x = pile$). $S_x$ is the surface over which subductions, continents or chemical piles can be found, and $\lambda$ is the latitude. Figure 5 shows the evolution of latitudes RMS in the simulation frame and the wandering frame. The subductions, continental, and chemical piles areas are defined the

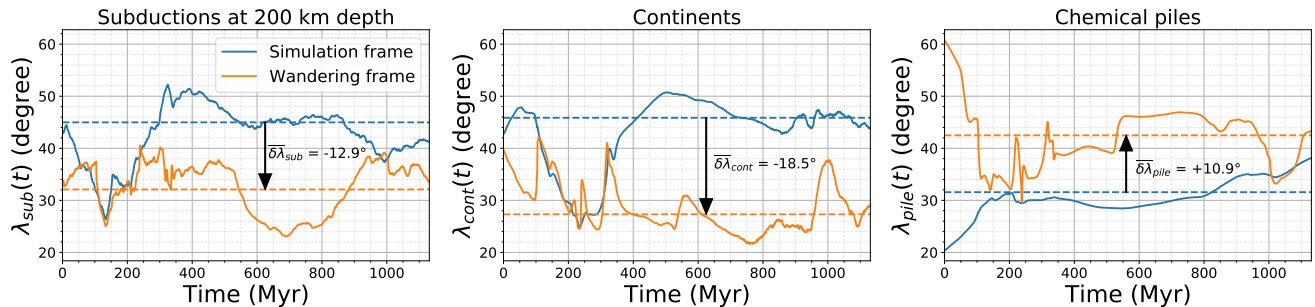

**Figure 5.** Time-evolution of the latitudes RMS of subducted slabs at 200 km, the continents, and the chemical piles, in the simulation frame (in blue) and the wandering frame (in orange). The dashed lines give the mean values between 300 Myr and the end of the simulation, excluding the initial 300 Myr long relaxation period.

same way as for the computation of the mean geoid anomaly shown in Fig. 4. The time-averaged values $\lambda_x$ are shown as the horizontal dashed lines. They were computed by excluding the first 300 Myr, which are dominated by the relaxation of the initial state. The differences $\overline{\delta\lambda_x}$ between the averaged values in the two frames are given in each case. As expected, subduction zones and continents tend to be at lower latitudes in the wandering frame. After the relaxation period, the RMS latitudes of subduction zones and continents are reduced by 12.9° and 18.5°, respectively, compared to the simulation frame. The latitude shift is stronger for continents, which create broader geoid anomalies than the subducted slabs. The latitudinal distribution of chemical piles is also affected by TPW. The piles are at higher latitudes in the wandering frame for most of the simulation. The difference between the mean latitude RMS in the wandering frame and in the simulation frame is 10.9°. The negative latitude shifts for subduction zones and continents are consistent with the positive geoid anomalies shown in Fig. 4. TPW keeps geoid highs away from the rotation axis, the continents and subduction zones are thus locked at low latitudes. The positive latitude shifts for chemical piles is also consistent with the mainly negative geoid anomalies observed over chemical piles. This moderate correlation between the geoid and the piles can be explained by the repulsive effect of subducting slabs in the lower mantle. The piles are pushed by the slabs, leading to an anti-correlation between the positions of slabs and piles. This





effect can explain why geoid highs can be found outside the piles, and thus the tendency of the piles to be at higher latitudes in the wandering frame than in the simulation frame.

The time-evolution of the TPW rotation rate is displayed in Fig. 6. TPW rotation rate never exceeds $5°$ Myr$^{-1}$, except

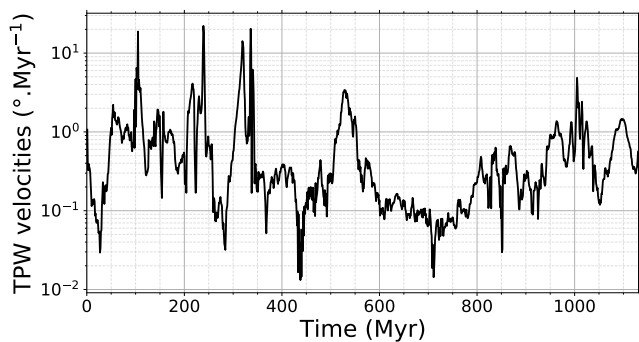

**Figure 6.** Time-evolution of the TPW rotation rate (in degrees per million years) in the simulation.

in the initial relaxation period, with four events above $10°$ Myr$^{-1}$ in the first 350 Myr. Our computation of TPW does not account for the delay due to the viscous adjustment of the equatorial bulge. This effect is expected to filter out the high-frequency displacement of the pole and to delay the alignment of the rotation axis to its new position (Cambiotti et al., 2011). A convenient way to approximate this delay is to apply a low-pass filter on the time-record of the position of the axis of maximum inertia. Applying a low-pass filter with characteristic times of 3 Myr or 10 Myr, as proposed by Cambiotti et al.

(2011), reduces the high frequency oscillations of the TPW rotation rate record visible in Fig. 6, but does not significantly affect the results of the PCA presented next.

### 3.3   PCA results

The patterns of the three first PCA components are displayed in Fig. 7a-c. Those patterns are perturbations from the average pattern, represented in Fig. 7d. The signs of the patterns are arbitrary, as the contribution of a PCA component to the total heat

flux is given by the product of the pattern with the amplitude, which changes sign with time.

The average heat flux pattern and the pattern of the first PCA component have very similar geometries. The average pattern is dominated by two mostly equatorial heat flux patches, surrounded by lower heat flux areas. The heat flux anomalies in the first PCA component pattern have the same geometry, though the signs of these anomalies are arbitrary. For a long period between 540 Myr and 970 Myr, the amplitude of the first component is positive. This corresponds to a period of stable equatorial

subduction zones leading to the formation of an equatorial supercontinent at around 700 Myr. Topography maps in Fig. 8 are shown for several snapshots to illustrate this supercontinent's formation and dispersal after 950 Myr. The stability of the subduction zones implies the stability of the chemical piles, and thus also the stability of the large scale CMB heat flux geometry. The dense piles indeed insulate the CMB, resulting in a lower mean CMB heat flux below them ($\approx 13$ mW m$^{-2}$)



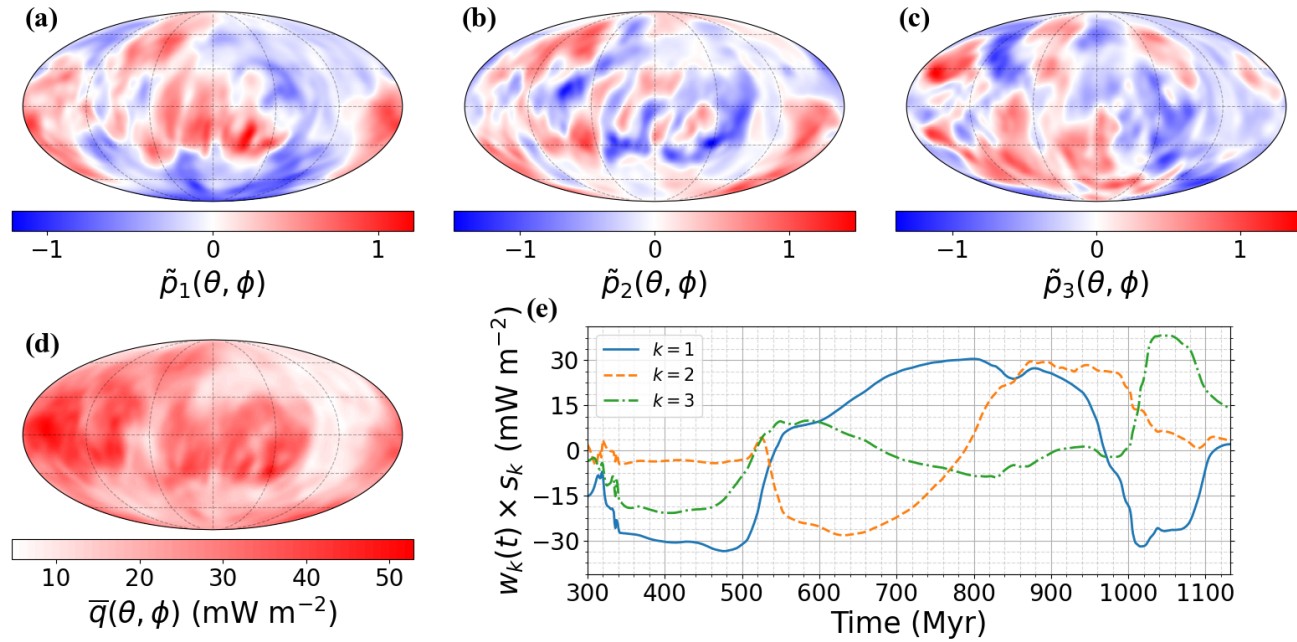

**Figure 7. (a)-(c)**: Patterns of the three first PCA components of the CMB heat flux. **(d)**: average pattern of the CMB heat flux. **(e)**: Time-evolution of $A_k(t) = w_k(t) \times s_k$ for the three first components. The PCA is computed on the CMB heat flux between 300 Myr and the end of the simulation at 1131 Myr, excluding the initial 300 Myr long relaxation period.

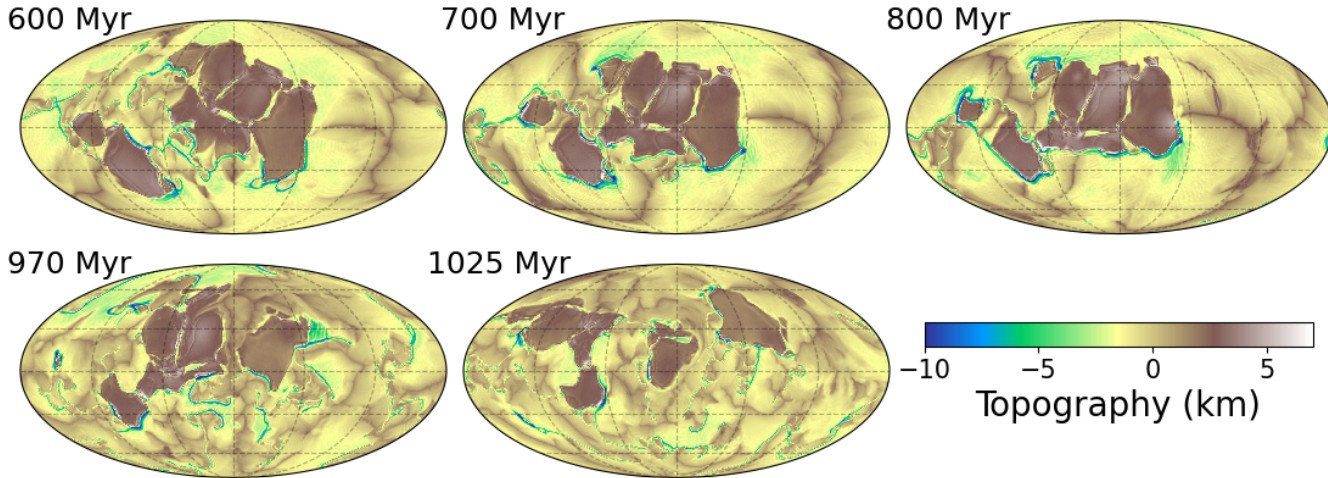

**Figure 8.** Topography of several snapshots showing the formation and dispersal of the supercontinent between 600 Myr and 1025 Myr.





than outside ($\approx 39$ mW m$^{-2}$). The snapshots at 600 Myr and 850 Myr thus have very similar continents, piles, and heat flux
distributions. The first component of the PCA can be seen as the large-scale effect of the supercontinent formation on the
chemical pile distribution and the CMB heat flux. The amplitude of the second component is also large for most of this period,
but with a change in sign in the middle of the period. This component can be seen as a perturbation to the mean pattern and
the first component, to adjust the heat flux to the slowly drifting chemical piles between 600 Myr and 900 Myr. The third
component has a smaller amplitude for the whole simulation, except during a short period between 1020 Myr and 1080 Myr,
right after the dispersion of the supercontinent. This pattern thus mostly reflects the effect of continental dispersion on CMB
heat flux.

The importance of the $k^{th}$ PCA component is quantified by the score $s_k$ which gives the square root of the variance explained
by this component. The variance explained by the $K'$ first components, written $\sigma_{K'}$, is thus given by

$$\sigma_{K'} = \sum_{i=1}^{K'} s_i^2. \tag{6}$$

Figure 9 gives the proportion of explained variance $\sigma_{K'}/\sigma_K$ as a function of $K'$, $\sigma_K$ being the total amount of variance in the
data set. This proportion rises quickly, with 95% and 99% of the variance being explained by the 28 and 55 first components,
respectively. This enables to significantly reduce the amount of stored data by considering only the first patterns, while keeping
most of the information.

Figure 10a gives the spherical harmonic power spectra of the PCA patterns for the 300 first PCA components. The dominant
spherical harmonic degrees of the patterns tend to increase with the PCA component number $k$ until it reaches $l_{max} = 50$ for
$k \sim 180$. The spectra then becomes more noisy for higher component numbers. The time spectra of the PCA weights in Fig.
10b show a similar structure compared to the spherical harmonic spectrum. The first PCA components are dominated by low
frequencies, and the dominant frequencies increase with the component number.

## 4 Discussion

### 4.1 Behaviour of the TPW

TPW is mostly governed by upper mantle subducting slabs and continents in the model. Geoid highs above subduction zones
are observed on Earth (Crough and Jurdy, 1980; Hager, 1984), and are thus expected in a mantle convection model that
reproduces plate tectonics. Geoid highs above upper mantle subduction zones have already been observed in a global mantle
convection model by Mao and Zhong (2021), with the difference that they prescribed plate velocities in their models. The
contrast between oceanic and continental areas has, however, no long-wavelength signal in the observed geoid (Crough and
Jurdy, 1980). This correlation between continents and geoid highs thus represents a first limit of our model in reproducing an
Earth-like geoid. It should be noted that continents are surrounded by subduction zones in the model, making it difficult to
separate the geoid signal arising from continents and subduction zones.





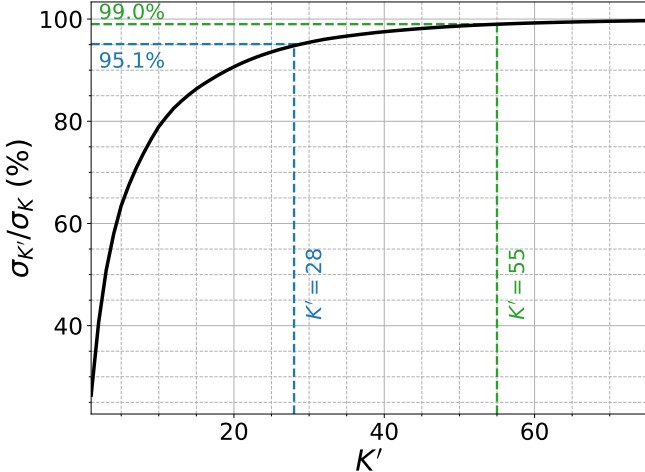

**Figure 9.** Variance explained by the $K'$ first PCA components $\sigma_{K'}/\sigma_K$, as a function of $K'$ ($\sigma_K$ is the total variance of the data set, $K$ being the total number of components). The amount of explained variance reaches 95% for $K' = 28$ and 99% for $K' = 55$.

The geoid highs observed above the Earth's LLSVPs (Dziewonski et al., 1977) are not reproduced by the model. On the
contrary, the piles are mostly associated with negative geoid anomalies. This could be due to an approximate modeling of
the LLSVPs, which could be dynamically buoyant plumes (Davaille and Romanowicz, 2020) rather than stagnant piles as
in the model. It is also possible that the very large geoid anomalies associated with subductions and continents are hiding a
potential signal from the piles. Positive geoid anomalies have been reproduced above non-buoyant piles by Liu and Zhong
(2015) in a long-wavelength mantle convection model. This model however does not reproduce plate-like behaviour, and thus
lacks subduction zones and continents which dominate the geoid produced by our model. Moreover, the geoid computed by
Liu and Zhong (2015) is not sensitive to the lower part of the mantle because of a compensation of the negative contribution of
the chemical piles to the geoid by a positive contribution of the upwellings above the piles. This could explain why the upper
mantle seems to control the geoid in our model. The geoid in their model is strongly correlated to the dynamic topography,
with geoid highs above large scale upwellings found above the piles. A similar effect could exist in our model, but it would be
hidden by contributions from subducting slabs and continents.

Phillips et al. (2009) found that supercontinent assembly should happen near the equator above a cold mantle, followed by
large TPW velocities during the dispersal. This is similar to what is observed in our model with the assembly of a supercontinent
around 700 Myr due to large-scale equatorial subductions, and a faster TPW during the dispersal around 1000 Myr.

The kilometer-sized amplitude of the geoid anomalies represents another limit of the computed geoid. The model has not
been adjusted to produce an Earth-like geoid anomalies amplitude, but it is probable that an adjustment of the physical pa-
rameters would allow for smaller geoid anomalies. This large amplitude could notably come from a lack of compensation
in the upper mantle because of the absence of phase changes. This inconsistency with the observed geoid however does not
affect our TPW implementation, which only depends on the geoid pattern. The amplitudes of the geoid anomalies are related





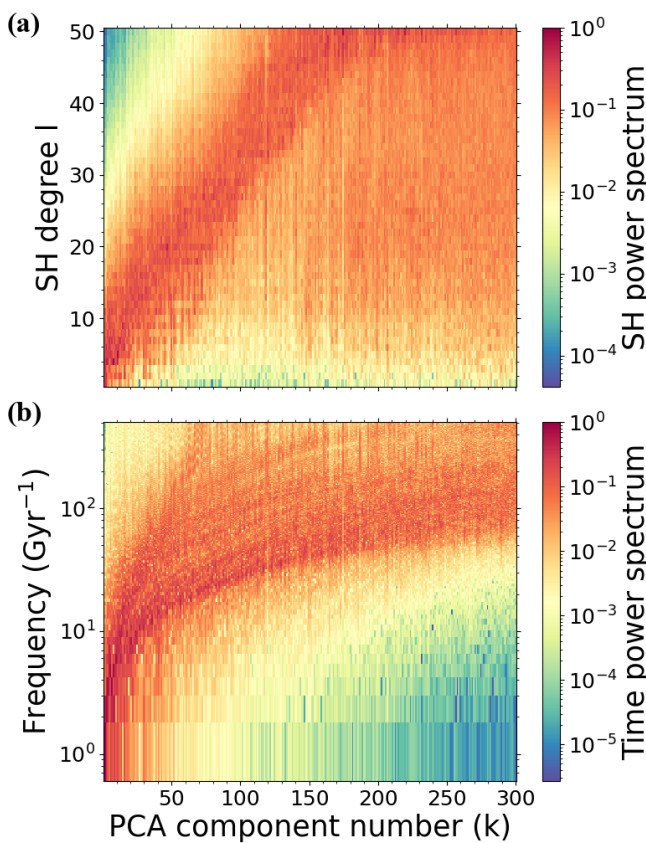

**Figure 10. (a)**: Spherical harmonic power spectrum of the PCA patterns $\tilde{p}_k(\theta, \phi)$ for the first 300 PCA components. **(b)**: Time power spectrum of the PCA weights $w_k(t)$ for the first 300 PCA components. The spectra are normalized to their maximum values.

to the inertia moments, which are in theory related to the viscous delay between the main inertia axis displacement and the
rotation axis displacement (Cambiotti et al., 2011). Our TPW implementation assumes no viscous delay, which removes the
dependency of the TPW on the geoid amplitudes. The TPW velocities are thus only controlled by the convection vigor in the
mantle. The TPW velocities range between $0.01°\,\mathrm{Myr}^{-1}$ and $1°\,\mathrm{Myr}^{-1}$ for most of the simulation. This range fits well with the
observed present-day TPW velocity of about $1°\,\mathrm{Myr}^{-1}$ (Gross and Vondrák, 1999) as well as the estimations for TPW events
reaching $\sim 0.3°\,\mathrm{Myr}^{-1}$ during the past 200 Myr (Besse and Courtillot, 2002). It is also in agreement with the TPW velocities
$\leq 2.5°\,\mathrm{Myr}^{-1}$ obtained by Phillips et al. (2009) in their models including rigid spherical continents, and with the theoretical
TPW velocity maximum limit of around $2.4°\,\mathrm{Myr}^{-1}$ determined by Tsai and Stevenson (2007) for the present-day mantle.
TPW velocities higher than several degrees per Myr could have existed during the Ediacaran period (Mitchell et al., 2011), and
velocities as high as $10°\,\mathrm{Myr}^{-1}$ have been modeled in cases of interchange between the maximum and intermediate inertia axis
(Inertia Interchange True Polar Wander) (Greff-Lefftz and Besse, 2014). The modeled rotation rate due to TPW thus broadly
agrees with expected and observed values for the Earth. As explained previously, our TPW implementation does not depend





on the amplitude of the inertia moments. The TPW velocities thus only depend on the convection vigor in the model, which has been adjusted to match the Earth's convection vigor. These Earth-like TPW velocity values are thus expected in our model.

### 4.2 LLSVPs thermal insulation effect, equatorial heat flux, and supercontinent formation

Chemical piles remain stable for a long time compared to the mantle convection timescale. The material below the piles thus
heats up, leading to a lower than average CMB heat flux below the piles. Outside of the piles, the slabs cool the lower mantle, yielding a high heat flux. The large scales of the CMB heat flux are thus controlled by a combined effect of subduction and chemical piles. High temperatures below the chemical piles are found in other thermochemical mantle convection models with prescribed plate velocities (Zhang and Zhong, 2011; Müller et al., 2022) and without prescribed plate velocities (Liu and Zhong, 2015). In our model, TPW maintains slabs at low latitudes. More than half of the model's duration (after removal
of the first 300 Myr) consists of a very stable period of equatorial subductions, leading to the formation of a supercontinent which lasts about 250 Myr. This creates strong positive equatorial heat flux anomalies, surrounded by hotter chemical piles. The averaged heat flux pattern as well as the first components of the PCA mostly reflect this period. The average pattern and the first component can be interpreted as a mean effect of this equatorial subduction period on the CMB heat flux. The second component accommodates the slow displacements of the piles during this period, while the third component mostly
corresponds to the results of the supercontinent dispersal on the CMB heat flux.

The CMB heat flux could have a significant impact on the geodynamo and the magnetic field reversal rate (Glatzmaier et al., 1999; Kutzner and Christensen, 2004; Olson et al., 2010). The existence of positive heat flux anomalies at the equator is notably found to largely increase the reversal frequency. The CMB heat flux patterns from the presented mantle convection model could thus strongly affect the reversal behavior of the geodynamo.

## 5  Conclusions

In this study, we explored the True Polar Wander (TPW) and the core-mantle boundary (CMB) heat flux produced by a mantle convection simulation (Coltice et al., 2019). The mantle model reproduces a plate-like behavior and includes basal chemical piles. We have thus considered TPW and CMB heat flux in relation to these features. The model produces very large geoid anomalies (about 20 times the Earth's geoid anomalies, see Fig. 2), dominated by the positive anomalies arising from continents
and subducting slabs. It results in mostly equatorial continents and subduction zones, while the chemical piles tend to stay away from the equator due to the repulsive effect of equatorial subducting slabs. Except for the effect of upper mantle subduction, this behavior does not match the observations regarding correlations between the geoid and the continents or chemical piles on Earth (Crough and Jurdy, 1980; Hager, 1984; Dziewonski et al., 1977). This work illustrates the difficulty of producing an Earth's like geoid in mantle convection models, despite including a realistic plate-like behavior.
Despite the large geoid anomalies, the displacement velocities of the rotation axis due to TPW shown Fig1 6 are in good agreement with the expected values for the Earth. The TPW velocities stay below several degrees per Myr for most of the simulation. Higher TPW velocities are found in the first 350 Myr of the simulation, notably due to the initial relaxation period



in the mantle convection model. These high velocity peaks up to 20° Myr$^{-1}$ are unrealistic for the present-day mantle (Tsai and Stevenson, 2007), but could have been reached in the past (Mitchell et al., 2011; Greff-Lefftz and Besse, 2014).

We computed a Principal Component Analysis (PCA) of the CMB heat flux to obtain the dominant heat flux patterns in the model. The most dominant patterns shown in Fig. 7 reflect the large-scale plate tectonics, notably the supercontinent cycle captured by the model. The CMB heat flux is governed by the thermal insulating effect of the basal chemical piles, which are shaped by the subducting slabs. It is thus expected to have a CMB heat flux driven by surface tectonics. The PCA consequently offers CMB heat flux patterns representative of plate tectonics in the model, notably regarding the supercontinent cycle.

*Code and data availability.*    The data are available on zenodo (https://doi.org/10.5281/zenodo.7249512) and the analysis scripts will be made available upon publication.

*Author contributions.*    HCN and SL conceived the project, NC provided the data and computed some additional outputs from the original model, TF wrote all the analysis scripts, performed the calculations and wrote most of the paper. All authors contributed to the discussion of results and the writing of the final paper.

*Competing interests.*    The authors declare no competing interest.

*Acknowledgements.*    The SHTns library (Schaeffer, 2013) was used for spehrical harmonics transforms and rotations. We thank Nathanael Schaeffer for implementing the spherical harmonic rotations within SHTns. This work was supported by the French Agence Nationale de la Recherche under grant ANR-19-CE31-0019 (revEarth)



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
