# Peer review of "True polar wander and heat flux patterns at the core-mantle boundary in a mantle convection simulation"

_EGUsphere, 2022_

## Referee Comment (RC2)

REVIEW OF: True polar wander and heat flux patterns at the core-mantle boundary in a mantle convection simulation

By Thomas Frasson[1], Stéphane Labrosse[2], Henri-Claude Nataf[1], and Nicolas Coltice[3]

This study explores a mantle convection simulation for just over 1 billion years with compositional differences in continents and chemical piles and plate-like behavior, the simulation is left to convect without imposed plate motion – enabling self-consistent physics. The authors investigate how true polar wander (TPW) and CMB heat flux change over this time and how they correlate with the 3 major characteristics of the plate tectonic/mantle system: continents, chemical piles and subduction zones. For TPW chemical piles seem to have little influence, while for heat flux, the piles are seen to produce areas of low CMB heat flux.

The study is a good contribution to understanding how plate-like behavior can drive deep forces within the earth related to rotation and CMB heat flux. I have some comments that warrant attention in the form of perhaps re-phrasing the work near the start. These involve perhaps rewriting the introduction, but the rest are minor. Very interesting and useful work. Thank you.

COMMENTS

One major comment I have about this work is that you claim to obtain some statistically realistic information concerning mantle convection parameters (lines 54-55). I do agree that without invoking surface velocities, dynamically, your model is self-consistent. However, by running only one simulation, can you really claim to have captured enough information to provide complete statistics?

In this vein, I think the introduction can be restructured. There is a lot of discussion about various wide-ranging themes in mantle dynamics. However, I believe that this work could be motivated in a different way. It's clear that you are recycling a very expensive computation from one of the co-authors. That is fine – and good(!) – but perhaps a better way to describe the manuscript would be to first state that this work is a follow up of Coltice's study. Explain the main finds of that study, but also what was missed in that study. What issues were not given enough attention in that study and what subjects warrant revisitings? Then you can explain why they are worth exploring with the text you describe about TPW and CMB heat flux. I think this would be a better way to present the paper.

MINOR COMMENTS

Line 25:

Reword:

"It is therefore important to evaluate what could be the evolution of the CMB heat flux on geologic timescales, in order to assess the consequences for the geodynamo."

To:

"It is therefore important to evaluate what the evolution of the CMB heat flux on geologic timescales could be, in order to assess the consequences for the geodynamo."

Line 26:

First, reword:

"The viscosity in the Earth's mantle being much larger than in the Earth's outer core, the mantle sees the CMB as an isothermal boundary, while the core sees the CMB as an imposed laterally varying heat flux."

To:

"Since the viscosity in the Earth's mantle is much larger than in the Earth's outer core, the mantle sees the CMB as an isothermal boundary, while the core sees the CMB as an imposed laterally varying heat flux."

However, I'm not sure I understand that the mantle sees the CMB as an isothermal boundary. Lateral temperature variations cause upwellings that arise from the CMB. Perhaps you mean something else?

Lines 88:
Briefly explain what is causing the density deficit and excess for the continents and chemical piles, respectively, and provide a citation.

Lines 108:
Remove "the" from "the True Polar Wander" and also the un-capitalize "True Polar Wander"

Line 112:
Change "capacity" to "capability"

Lines 131–135:

Perhaps explain this clearer. Remove some pronouns and use proper nouns when referring to axes. It's hard to follow, even though it's a relatively simple concept.

Figure 5:

Interesting discussion on this. Is it possible to plot, rather than RMS over time, a whole histogram with time? I'm envisioning a figure where the y-axis is the latitude. At each time (along the x-axis) there is a vertical histogram, where you could use color intensity to show the distribution. It's a lot more informative and will highlight the point about the chemical piles being aligned over a wide range of latitudes.

Line 359:

Typo in "Fig1 6"?